# Quantum Biochemistry and MM-PBSA Description of the ZIKV NS2B-NS3 Protease: Insights into the Binding Interactions beyond the Catalytic Triad Pocket

**DOI:** 10.3390/ijms231710088

**Published:** 2022-09-03

**Authors:** Valdir Ferreira de Paula Junior, Mauricio Fraga van Tilburg, Pablo Abreu Morais, Francisco Franciné Maia Júnior, Elza Gadelha Lima, Victor Tabosa dos Santos Oliveira, Maria Izabel Florindo Guedes, Ewerton Wagner Santos Caetano, Valder Nogueira Freire

**Affiliations:** 1Biotechnology & Molecular Biology Laboratory, State University of Ceará, Fortaleza 60714-903, Brazil; 2Federal Institute of Education, Science and Technology of Ceará, Campus Horizonte, Horizonte 62884-105, Brazil; 3Departamento de Ciências Naturais, Matemática e Estatística, Universidade Federal Rural do Semi-Árido, Mossoró 59625-900, Brazil; 4Laboratório Central de Saúde Pública do Ceará (LACEN), Fortaleza 60120-002, Brazil; 5Federal Institute of Education, Science and Technology of Ceará, Campus Fortaleza, Fortaleza 60040-531, Brazil; 6Department of Physics, Federal University of Ceará, Fortaleza 60455-760, Brazil

**Keywords:** quantum biochemistry, MM-PBSA, oxyanion orifice, NS2B-NS3, Zika virus

## Abstract

The Zika virus protease NS2B-NS3 has a binding site formed with the participation of a H51-D75-S135 triad presenting two forms, active and inactive. Studies suggest that the inactive conformation is a good target for the design of inhibitors. In this paper, we evaluated the co-crystallized structures of the protease with the inhibitors benzoic acid (5YOD) and benzimidazole-1-ylmethanol (5H4I). We applied a protocol consisting of two steps: first, classical molecular mechanics energy minimization followed by classical molecular dynamics were performed, obtaining stabilized molecular geometries; second, the optimized/relaxed geometries were used in quantum biochemistry and molecular mechanics/Poisson–Boltzmann surface area (MM-PBSA) calculations to estimate the ligand interactions with each amino acid residue of the binding pocket. We show that the quantum-level results identified essential residues for the stabilization of the 5YOD and 5H4I complexes after classical energy minimization, matching previously published experimental data. The same success, however, was not observed for the MM-PBSA simulations. The application of quantum biochemistry methods seems to be more promising for the design of novel inhibitors acting on NS2B-NS3.

## 1. Introduction

The World Health Organization (WHO) acknowledges the Zika virus (ZIKV) as a public health problem. ZIKV belongs to the Flaviviridae family, and is transmitted through the bite of female *Aedes aegypti* and *Aedes albopictus* mosquitoes. This pathogen was first identified in humans in 1952 in Uganda and the United Republic of Tanzania [1]. After the first reports, outbreaks were identified in the USA, Africa, Asia, and other regions of the world [2]. In 2015, Brazil suffered from a major Zika virus epidemic causing fetal microcephaly and brain damage in newborns, infection in pregnant women, and Guillain–Barré syndrome in adults [3,4].

The Zika virus genetic material consists of a ~10-kb single-stranded positive-sense RNA chain containing a 5′ untranslated region (UTR) with ~100 nucleotides (nt) and a single open reading frame of ~10 kb, and ~420-nt 3′ UTR [5]. It expresses a polyprotein with 3000 amino acids separated into structural (capsid, C; precursor membrane, prM; and envelope protein, E) and nonstructural (NS1, NS2A, NS2B, NS3, NS4A, NS4B, and NS5) [6,7,8,9] proteins. Viral replication functional complexes are formed in specific replication compartments of the endoplasmic reticulum (ER) with the participation of all seven proteins in the nonstructural group [10]. NS1 is translated inside the ER lumen [11,12], while NS2A, 2B, 4A, and 4B are integral components of the ER membrane [13,14,15].

The NS3 and NS5 enzymes stand out for their functionality: NS5 acts as a polymerase, synthesizing negative-sense RNA, which results in a double-strand RNA intermediate unwound by NS3. Both proteins cooperate in the methylation of the positive-sense RNA to produce more viral particles, and NS3 has a protease domain that contributes to the cleaving of the viral polyprotein. NS3 and NS5 remain in the cell cytosol without direct anchoring to the ER [16,17,18]. The viral protease is formed by the conjugation of the N-terminal portion of NS3 and the cofactor region of NS2B [19,20]. This complex forms a catalytic triad at the active site characterized by the presence of the H51, D75, and S135 residues [14,21]. Its oxyanion orifice exhibits two conformations mentioned in the literature. The open form plays an important role in the binding of the inhibitor to the substrate (see PDB: ID 5GJ4), isolating residues A132, G133, T134, and S135 from the NS3 subunit. It interacts with residues T127, G128, K129, and R130 from the NS2B subunit at positions P1 and P2, respectively [14,22], through hydrogen interactions [21] (Figure 1A). When the inhibitor benzimidazole-1-ylmethanol binds to NS3, on the other hand, a closed conformation of the oxyanion orifice is produced [14,23] (Figure 1B). Considering the superposition of the crystallized structures PDB: ID 5H4I and 5GJ4, a change from the active to the inactive state in the catalytic triad of the NS3 protease is observed (Figure 1C).

Inhibitor binding changes the orientation of the G133 and A132 fragments from the active to the inactive state [24] (Figure 2A). Unbound NS2B-NS3 is considered the appropriate model for the development of inhibitors. Experimental findings demonstrate co-crystallization of complexes in the inactive conformation of the NS2B-NS3 oxyanion orifice without specific interaction with the inhibitor [25,26], and stabilizing the protease conformation without direct interaction with NS2B (Figure 2B–E). The PDB files 5YOD (NS3 co-crystallized with the inhibitor benzoic acid, BC) and 5H4I (NS3 co-crystallized with the inhibitor benzimidazole-1-ylmethanol, B1Y) exhibit differences at positions 46 (5YOD, residue E; 5H4I, residue D), and 202 (5YOD, residue S; 5H4I, residue C) (Figure 2F).

Computational methods can be used to study protein–ligand and protein–protein interactions, predicting the binding mode of an inhibitor to a substrate [27,28]. These methods can incorporate different levels of theoretical principles. Approaches incorporating classical and/or quantum mechanics allow one to estimate the conformation of a ligand in a target catalytic pocket to evaluate steric conflicts, thermodynamic variations, geometric alteration, and energetic contributions to a specific complex [29,30,31]. In a hybrid quantum mechanics/molecular mechanics (QM/MM) strategy, quantum or ab initio methods such as Hartree–Fock (HF) or density functional theory (DFT) are applied to “fragments” of the systems under study, making it possible to analyze quantitatively the properties of the protein–ligand system [32].

Several fragmentation models have been proposed in the last few years, such as the molecular fractionation with conjugated caps (MFCC) model, the fragment molecular orbital (FMO) method, the molecular tailoring approach (MTA), and the generalized energy-based fragmentation method (GEBF) [33]. In particular, the MFCC method combined with DFT calculations provides a very practical and helpful way to evaluate the interaction energies in protein–ligand and protein–protein complexes [34,35]. Additionally, the molecular mechanics/Poisson–Boltzmann surface area (MM-PBSA) model is considered a powerful tool to estimate the interactions exerted on receptor–ligand, protein–protein, and protein–nucleic acid complexes [36]. This method consists of using snapshots of molecular dynamics (MD) to calculate the free energy of binding. Evidently, MM-PBSA does not require many computing resources, and the calculations for estimation of free energy of binding are fast. However, the use of this tool is not free, making it restricted to certain users. On the other hand, the g_mmpbsa tool is available to all users, and maintained by the open-source drug discovery consortium (OSDD) [37].

In this study, we apply several techniques, such as docking, molecular dynamics, MFCC quantum biochemistry, and MM-PBSA, to investigate the influence of the amino acid residues H51, S135, and ASP75 on the recognition of inhibitors in the inactive conformation of the NS2B-NS3 oxyanion orifice. We characterize the energy contributions per amino acid residue to the complex formation, highlighting which ones are essential for the stabilization of the Zika virus protease, and aiming for the design of novel Zika virus protease inhibitors.

## 2. Results and Discussion

### 2.1. Molecular Docking

The most significant binding energy scores occur in the connection area S1 on the C-terminal main side comprising the positions P1, P1′-P2, reaching −4.09 kcal/mol, for the interaction of BC with the target NS2B-NS3 5YOD, and −4.74 kcal/mol for B1Y with 5H4I, as expected from the experimental data [21,26]. One must remark that molecular docking is a qualitative metric aiming to reproduce the chemical potentials of a geometric fit, determining the conformation of bound structures through a score ranking based on the free energy of binding.

### 2.2. Molecular Dynamics

Molecular dynamics allows the evaluation of the atomic movement under predetermined conditions through the definition of pre-analysis (energy minimization, solvation, temperature, and pressure) and post-analysis (root mean square deviation of atomic positions, RMSD, and root mean square fluctuation, RMSF) parameters. The alpha carbon (Cα) RMSD reveals that the 5YOD (md5/dft) carbon backbone coordinates diverge by about 0.2 nm relative to the crystallographic data oscillating with an amplitude less than 0.05 nm (Figure 3A). The RMSF plot for every residue in the protein structure shows little variation (less than 0.1 nm) for the residues in the oxyanion orifice (H51, D75, A132, G133, T134, and S135), suggesting a preserved active site (Figure 3B). The RMSD of the 5YOD complete structure reveals a tendency of stabilization at nearly 0.4 nm, intercalated with a series of drops to below 0.2 nm, with minimum values at about 1 ns, between 2 and 2.5 ns, and after 4.1 ns (Figure 3C). In the analysis of the dynamics of the 5YOD (md5) complex with BC, one can observe that the BC molecule was rotated by 180° about the co-crystallized structure (Figure 3D–F). This change of position can be caused by the repulsion between the N atoms of H51 (2.67 Å distant), the S135 oxygen atom (1.75 Å distant), and the inhibitor oxygen atom of BC, as seen in the co-crystallized structure.

The 5H4I (md5/dft) temporal evolution can be seen in Figure 4A. The Cα RMSF plot for the relevant residues (H51, D75, A132, G133, T134, and S135) also reveals the preservation of the active site (Figure 4B), while the Cα RMSD stabilizes around 0.3 nm (Figure 4A). The simulation of the B1Y inhibitor shows an RMSD increase from 0.1 nm to about 0.4 nm in the time interval of 2 ns, followed by stabilization just above 0.4 nm, being absent the RMSD plunges observed for the full BC:5YOD complex (Figure 4C). In general, we observed that the conformation adopted by B1Y after the MD run does not interact considerably with any amino acid of the catalytic triad (H51, D75, and S135). The relative change of position of B1Y at the binding site is evident by observing the overlap in Figure 4D–F.

Figure 5 shows 2D diagrams revealing the most important interactions of both BC and B1Y at the binding site of the NS2B-NS3 protease from molecular dynamics (md5) and molecular energy minimization (em). One can see, in Figure 5A,B, that due to the rotation of BC, a new hydrogen bond between O1 (BC) and the residue Y130 was established in the structure found after the molecular dynamics procedure (distance between molecular centroids (d_c_) = 6.45 Å, hydrogen bond length (hbl) = 2.4 Å). A hydrogen bond with G151 (d_c_ = 4.14 Å, hbl = 3.3 Å) present in the molecular energy minimization is lost. B1Y, on the other hand, changes from an em conformation with a hydrogen bond to Y150 (d_c_ = 4.93 Å, hbl = 1.8 Å) (Figure D) to an md5 conformation, in which hydrogen bonds to I156 (d_c_ = 5.17 Å, hbl = 2.7 Å) and V126 (d_c_ = 4.09 Å, hbl = 2.1 Å) are formed (Figure C).

The variation of the positions of the inhibitors (BC and B1Y) in the catalytic site is due to the presence of repulsion points that displace the inhibitor. This binding conformation change can have a significant influence on the displacement of the B1Y inhibitor, and the presence of repulsive regions can reinforce this effect. Examples of repulsive regions can be seen in the co-crystallized structure. One can pinpoint, for example, a hydrogen atom of Y150, an oxygen atom of B1Y (4.93 Å distant), a hydrogen atom (H) of I156 (5.17 Å), and V126 relative to a B1Y oxygen atom (3.14 Å) (see Figure 5C,D). In MD simulations, it is a well-known fact that deviations from the co-crystallized geometry for small ligand molecules at the active site are typical [38].

### 2.3. Quantum Biochemistry Simulations

Understanding the energetic interplays involved in the stabilization of a protein–ligand complex is crucial in the development of new drugs [39]. The quantum analysis of the interactions is very helpful to this end. Here we use the em and md5 structures to evaluate the binding energy profile of BC and B1Y estimating their total binding energies, as shown in Figure 6. The total binding energy for a distance r from the ligand is obtained by adding up the ligand–residue interaction energies evaluated through the MFCC approach. Only the residues within r are included in this sum. As r increases, the total interaction energy stabilizes and reaches a value related to the total binding energy of the ligand through a sign change.

The 5YOD total energy profile for the md5 structure (Figure 6A) exhibits a sharp decrease as the radius from the ligand increases to about 3.5 Å, reaching about −20 kcal/mol and keeping a constant value between 3.5 Å and 5.0 Å due to the absence of amino acid residues within this distance interval (one can see this by looking at the gap in the blue scatter pattern). For distances larger than 5 Å, there is a slight decrease in the total binding energy as more residues are considered, stabilizing the total energy at nearly −22 kcal/mol for a distance of 7 Å. In the case of the em structure (Figure 6C), the total energy decrease for small distance values is not so sharp, as some residues are contributing significantly to its value between 2 and 3 Å. There is a total energy plateau of about −15 kcal/mol starting at 4 Å and extending to 5.2 Å (another amino acid residue gap), and a small decrease between 5.2 and 5.3 Å with a secondary plateau of −16.5 kcal/mol between 5.3 and 7 Å. The total energy of the complex stabilizes at about −17 kcal/mol between 7 and 8 Å.

For the B1Y binding profile based on the 5H4I data, the total interaction energy of the structure obtained from the molecular dynamics last snapshot decreases rapidly as amino acids between 2 and 3 Å from the ligand are considered in its sum (Figure 6B), reaching about −29 kcal/mol. For distances between 3 and 4 Å, there is a small decrease to about −30 kcal/mol that does not change significantly for distances up to 5.6 Å. Total energy stabilization near −31 kcal/mol occurs for 7 Å. This behavior can be contrasted with the interaction energy profile obtained considering the complex after energy minimization (Figure 6D), where one finds a smoother energy decrease between 2 and 4 Å, reaching −24 kcal/mol, an energy plateau between 4.5 and 5.5 Å, and an additional decrease to −26 kcal/mol, a value that remains stable for distances up to 8 Å. One can say that the molecular dynamics simulation leads to geometries with smaller total interaction energies than the energy minimization procedure, and that the main residues responsible for the binding of the ligands are within 4 Å from them. Lastly, the binding energies found for the four structures when r = 8 Å were: 5YOD (md5/dft) 22.22 kcal/mol, 5YOD (em/dft) 16.83 kcal/mol, 5H4I (md5/dft) 30.92 kcal/mol, and 5H4I (em/dft) 29.26 kcal/mol.

A detailed evaluation of the amino acid residues involved in the complexation of BC and B1Y to the NS2B-NS3 protease is shown in Figure 6. It depicts BIRD (binding site, interaction energy, and residue domain) panels revealing all residues present in the binding pocket up to 8 Å. The variables used to portray the mechanisms involved in the protein–ligand complex are the residue, its interaction energy, and the distance of interaction with the ligand. The energetic importance of the catalytic triad formed by the residues H51, D75, and S135, and the determination of other residues essential for the stabilization of the complex, can be easily checked through the quantum-level results. For the 5YOD (md5/dft) system, the binding of BC to the amino acid residues Y130 (2.13 Å), Y150 (2.19 Å), A132 (2.31 Å), S135 (2.35 Å), P131 (2.43 Å), Y161 (2.79 Å), H51 (3.07 Å), G151 (3.26 Å), T134 (3.36 Å), and D129 (4.55 Å) must be highlighted (Figure 7A). In the stabilization of 5YOD (em/dft), we consider the close residues S135 (1.98 Å), G151 (2.24 Å), Y130 (2.59 Å), H51 (2.75 Å), A132 (2.78 Å), Y150 (3.15 Å), Y161 (3.17 Å), D129 (3.63 Å), N152 (3.81 Å), and T134 (3.98 Å) as the most relevant (Figure 7C).

For the 5H4I (md5/dft) system, we identify the residues D129 (2.03 Å), V154 (2.10 Å), V126 (2.13 Å), V155 (2.39 Å), G153 (2.47 Å), I156 (2.67 Å), A125 (2.79 Å), F116 (2.79 Å), K157 (2.81 Å), Y150 (2.93 Å), A127 (3.01 Å), G151 (3.28 Å), L128 (3.96 Å), and Y130 (4.04 Å) interacting more intensely with B1Y (Figure 7B). On the other hand, for the 5H4I (em/dft) complex, we assign the role of the residues Y150 (1.78 Å), Y161 (2.15 Å), Y130 (2.36 Å), D129 (2.60 Å), G151 (2.61 Å), S135 (2.95 Å), A132 (2.96 Å), P131 (3.12 Å), T134 (3.35 Å), V155 (3.76 Å), S163 (4.23 Å), N152 (4.72 Å), and V162 (5.71 Å) as interacting with B1Y (Figure 7D).

In the evaluation of the four systems, 5H4I (md5/dft), 5H4I (em/dft), 5YOD (md5/dft), and 5YOD (em/dft), we observe that both the 5H4I (em/dft) and 5YOD (em/dft) complexes must portray the energy contributions consistent with the experimental findings [22,26], as they more closely follow the crystallographic data. Two residues of the catalytic triad, S135 and H51, have, respectively, interaction energies of −0.98 kcal/mol and 0.03 kcal/mol for 5H4I (em/dft). There is a much larger energy contribution involving the participation of the Y161 (−7.71 kcal/mol), Y150 (−5.25 kcal/mol), and A132 (−3.32 kcal/mol), in agreement with the experimental data from Zhang and collaborators, which describes the B1Y inhibitor positioned between the aromatic ring of Y161 and A132 through stacking interactions and forming a hydrogen bond with Y150 [22], (a significant electron density leads to an π–π stacking interaction with Y161). Along with the residues already mentioned, we also observed the participation of V155 (−1.97 kcal/mol).

Considering the 5YOD (em/dft) system, the energetic contributions of the amino acid residues belonging to the catalytic triad were only −0.69 kcal/mol (S135), −0.55 kcal/mol (H51), and −0.07 kcal/mol (D75). In contrast, there are much more significant interaction energies with the amino acids Y161 (−3.49 kcal/mol) and A132 (−3.89 kcal/mol), in agreement with Li et al. [26], who pointed out that the same residues participate in the formation of π–π stacking interactions. We also note the participation of Y150 (−1.23 kcal/mol) and G151 (−3.33 kcal/mol) with relevant interactions.

### 2.4. MM-PBSA

The MM-PBSA method usually produces intermediate-quality results as it ignores the conformational changes of the target and the ligand in the complex and the entropy of the water molecules at the active site, with system-dependent performance [40]. However, MM-PBSA calculations for a minimum set of minimized instances can provide some insight into the composition of the total binding energy [41,42]. The calculated free energies of binding for the minimized complexes were: −5.01 kcal/mol (5H4I (em/pbsa)), −5.28 kcal/mol (5YOD (em/pbsa)), −7.10 kcal/mol (5H4I (md5/pbsa)), and −8.54 kcal/mol (5YOD (md5/pbsa)), as shown in Figure 8. The significant difference between the MM-PBSA binding energy values and the previous quantum-level predictions follows a pattern observed in other studies [43,44].

The binding energies of the catalytic triad H51-D75-S135 were evaluated through the MM-PBSA calculation. For the 5H4I (md5/pbsa) system, we obtained −0.0214 kcal/mol (H51), −0.342 kcal/mol (D75), and 0.0585 kcal/mol (S135). In the case of 5H4I (em/pbsa) systems, we have found 0.537 kcal/mol (H51), −0.131 kcal/mol (D75), and 0.0416 kcal/mol (S135). The complex 5YOD (md5/pbsa) has H51-D75-S1 interaction energy values of 0.452 kcal/mol, 0.0309 kcal/mol, and −0.200 kcal/mol, in respective order. For 5YOD (em/pbsa), the corresponding H51-D75-S1 values were 0.579 kcal/mol, −0.0674 kcal/mol, and 0.687 kcal/mol, respectively. The most significant energy contributions obtained using the MM-PBSA method are in two distinct regions for both complexes at positions 30aa–55aa and 129aa–165aa (see Figure 9). Other residues worth mentioning due to their energy contributions in comparison with the MFCC results are: in 5YOD (em/pbsa), Y161 with −0.552 kcal/mol and A132 with −0.626 kcal/mol; in 5H4I (em/pbsa), P131 with −1.028 kcal/mol, A132 with −1.032 kcal/mol, and Y161 with −1.754 kcal/mol; in 5YOD (md5/pbsa), A132 with −0.748 kcal/mol; in 5H4I (md5/pbsa), Y150 with −0.485 kcal/mol and V155 with −1.157 kcal/mol. Different from the quantum-level calculations, no new information could be inferred from the MM-PBSA calculations. The use of other approaches such as MM-PBSA, MM-GBSA, QM/MM-PBSA, LIE, and MM-PBSA may improve this picture [45,46].

## 3. Materials and Methods

### 3.1. Structural Analysis and Preparation

Preparation of systems. We used the X-ray structures of the Zika virus NS2B-NS3 protease complexed with its inhibitors benzoic acid (BC) and benzimidazole-1-ylmethanol (B1Y), downloaded under the codes 5YOD (1.90 Å resolution) [26] and 5H4I (2.00 Å resolution) [21] from the Protein Data Bank (https://www.rcsb.org/, accessed on 13 October 2021). Water molecules and ions were removed from the protease structures using the PyMOL software v2.5, creator Warren Lyford DeLano, California/EUA [47] and analyzed in SPDBViewer to fix the residues and any missing atoms.

Subsequently, fast energy minimization was performed using the CHARMM force field (http://spdbv.vital-it.ch/, accessed on 13 October 2021) [48]. Hydrogen atoms were adjusted according to their protonation states using the PDB2PQR server (http://agave.wustl.edu/pdb2pqr/, accessed on 13 October 2021) [49], resulting in a suitable protease structural model for the simulations. Finally, the ligands were optimized at pH 4.6 using Avogadro [50] with the added hydrogen atoms.

The characterization of the amino acid residues involved in the ligand–receptor interaction was performed on Accelrys Discovery Studio 2021 client. The following cutoff interaction distances were imposed: van der Waals: 0.70 Å; hydrogen: 3.4 Å (angle cutoff minimum of 90° and maximum of 180°); halogen: 3.70 Å (and minimum angle of 120° and maximum of 180°); sulfur/x: 1.00 Å (maximum angle of 90°); atom/π: 5.00 Å (maximum angle of 40°); π/π: 6.00 Å (minimum angle of 50°); alkyl: 5.50 Å. The spatial analysis of the residues was performed using PyMOL 2.0 [51,52].

### 3.2. Molecular Modeling

Rigid Body Docking. The NS2B-NS3 complexes from the 5YOD and 5H4I files were processed for molecular docking using AutoDockTools 1.5.6 [53]. For each complex, a grid box was created in the region of the active site delimiting the portion of the oxyanion orifice in the inactive configuration (closed) with coordinate grid centers (in Å) x = −9.543, y = 2.895, z = −15.683 for 5YOD (md5), and x = −9.543, y = 2.895, z = −15.683 for PDB: ID 5H4I (md5). These steps were performed using PyRx, and AutoDock Vina with default settings for exhaustiveness and energy range [54].

### 3.3. Molecular Dynamics

Molecular simulations in water. Simulations were performed to examine the stability of the predicted protein–ligand complexes using GROMACS version 5.1 with the CHARMM36 force field [55]. The solvation of the system was performed using the dodecahedron box with the TIP3P [56] water model and a distance to the edge of the box set at 2.0 nm. To neutralize the system, counterions were added (Na^+^ and Cl^−^) replacing the water molecules in each solvated system. The system was relaxed by energy minimization using the steepest descent energy minimization with 50,000 steps and a maximum force threshold of 1000 kJ mol^−1^ nm^−1^ to avoid stereochemical conflicts (these optimized structures will henceforth be labeled “em”). Afterward, the system underwent equilibrations lasting 100 ps in an NVT ensemble and 100 ps in an NPT ensemble [57]. The Berendsen (velocity-rescaled) thermostat [58] and Parrinello–Rahman barostat [59] were used. Covalent bonds were defined using the algorithm linear constraint solver (LINCS) [60], and the particle mesh method of Ewald (PME) was adopted for the treatment of electrostatic interactions [61]. Short-range electrostatic and van der Waals distance cutoffs were both set to 1.2 nm, with a coupling time of 100 ps [62]. The complexes 5YOD (em) and 5H4I (em) were subjected to a simulation time of 5 ns. The structures from the last frame of the 5 ns molecular dynamics simulations are labeled here as “md5”.

### 3.4. Quantum Biochemistry

We obtained all interaction energies involving residues of the protease of ZIKV within 8 Å from the ligand. Density functional theory (DFT) calculations were performed using the DMol3 [63] code adopting the generalized gradient approximation (GGA) function developed by Perdew, Burke, and Ernzerhof (PBE) [64], combined with the Tkatchenko and Scheffler (TS) [65] long-range dispersion correction. The DNP+ [66] basis set (double numerical plus polarization with the addition of diffuse functions) was chosen to expand the Kohn–Sham orbitals for all electrons. The orbital cutoff radius was set to 5.5 Å, and the total energy convergence threshold set to 10^−6^ Ha. These calculations were performed adopting the COSMO continuum solvation model [67] with dielectric constant ε = 40 for all residue–ligand interactions. This value was assumed considering the work of Morais et al. [68], which indicated that a dielectric constant value of around 40 reproduces the principal global and local effects of the dielectric response of protein complexes.

### 3.5. Thermodynamic Calculations

The MM-PBSA approach is a powerful strategy to evaluate the binding free energy in a protein–ligand system [42]. This study adopted the single path method (STM), avoiding structural deviations [69]. The g_mmpbsa tool can be used for high-yield MM-PBSA calculations, allowing one to calculate individual energy terms and the total interaction energy, performing energy decompositions for each residue through Python scripts [37].

The polar energy contribution, Gpolar, to transfer the solute from a continuous medium with a low dielectric constant (ε = 1) to a continuous medium with the dielectric constant of water (ε = 80) was calculated for a 0.5 Å upper grid spacing, a temperature of 296 K, and a salt concentration of 0.15 M. For the nonpolar contribution, Gnonpolar, the surface area accessible to the solvent (SASA) was considered. The complexes submitted to the MM-PBSA procedure are labeled 5YOD (em/pbsa), 5YOD (md5/pbsa), 5H4I (em/pbsa), and 5H4I (md5/pbsa).

## 4. Conclusions

The NS2B-NS3 protease is considered an attractive target for the design of Zika virus inhibitors. Its closed conformation stands out for providing valuable information on the energetic contributions involved in the recognition of small molecules. We have used here a consistent protocol performing classical molecular mechanics energy minimization/classical molecular dynamics simulation followed by quantum-level energy calculations to investigate the co-crystallized structures 5YOD (with benzoic acid, BC) and 5H4I (benzimidazole-1-ylmethanol, B1Y). Strong agreement was observed with experimental data. The quantum approach enabled the identification of important interacting residues: Y161 (−7.71 kcal/mol), Y150 (−5.25 kcal/mol), and A132 (−3.32 kcal/mol) for 5H4I (em/dft); Y161 (−3.49 kcal/mol), A132 (−3.89 kcal/mol), and G151 (−3.33 kcal/mol) for 5YOD (em/dft). The geometry obtained for the last frame of the classical molecular dynamics simulation presented even stronger binding energies at the quantum level. In contrast, it was observed that MM-PBSA simulations based on a single trajectory did not reproduce the experimental data, with small binding energy values. Both quantum- and MM-PBSA-level theories, however, predict tiny binding energies associated with the catalytic triad of residues H51-D75-S135. The quantum-level theory was shown to be more able to predict the binding energetics in contrast with available experimental data.

## Figures and Tables

**Figure 1 ijms-23-10088-f001:**
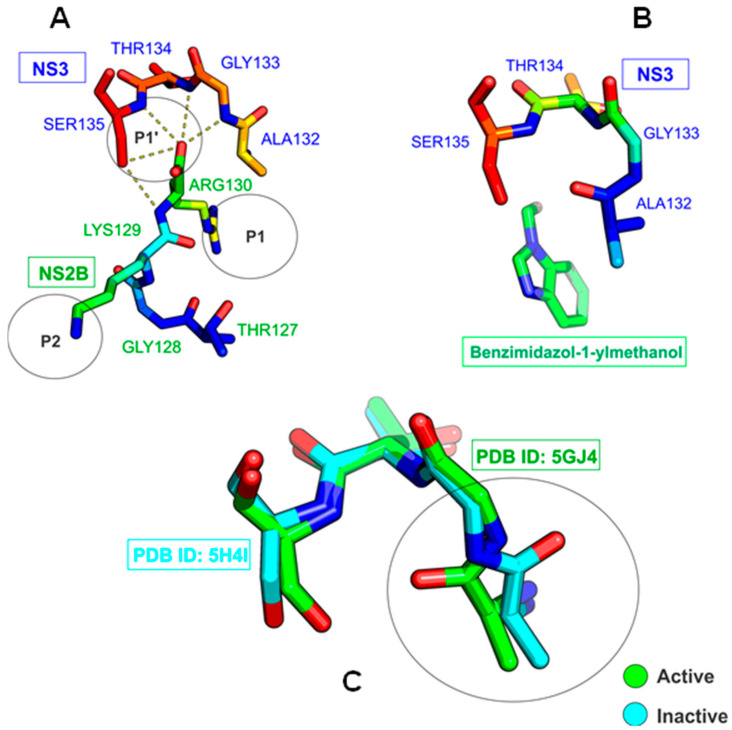
Representation of the oxyanion orifice of NS2B in the open state coupled to NS3 PDB ID: 5GJ4 (**A**), and in the closed state coupled with a PDB ID: 5H4I inhibitor (**B**). Overlapping of NS3 structures showing how the geometry of the oxyanion orifice changes (**C**).

**Figure 2 ijms-23-10088-f002:**
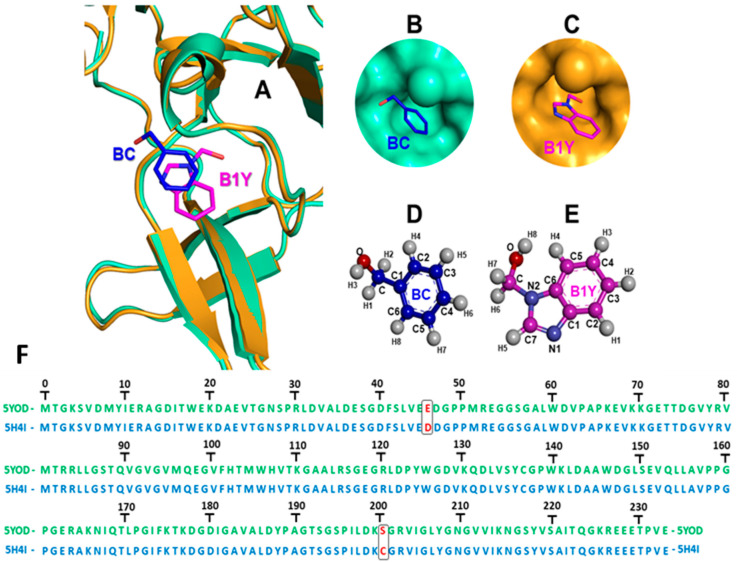
(**A**) Superimposition of the Zika virus protease co-crystallized with inhibitors benzoic acid (BC, 5YOD PDB file, emerald color) and benzimidazole-1-ylmethanol (B1Y, 5H4I PDB file, mustard color) at the active site; (**B**) BC in 5YOD; (**C**) B1Y in 5H4I; (**D**,**E**) 2D representation of BC and B1Y; (**F**) alignment of the viral proteases according to the 5YOD and 5H4I PDB files.

**Figure 3 ijms-23-10088-f003:**
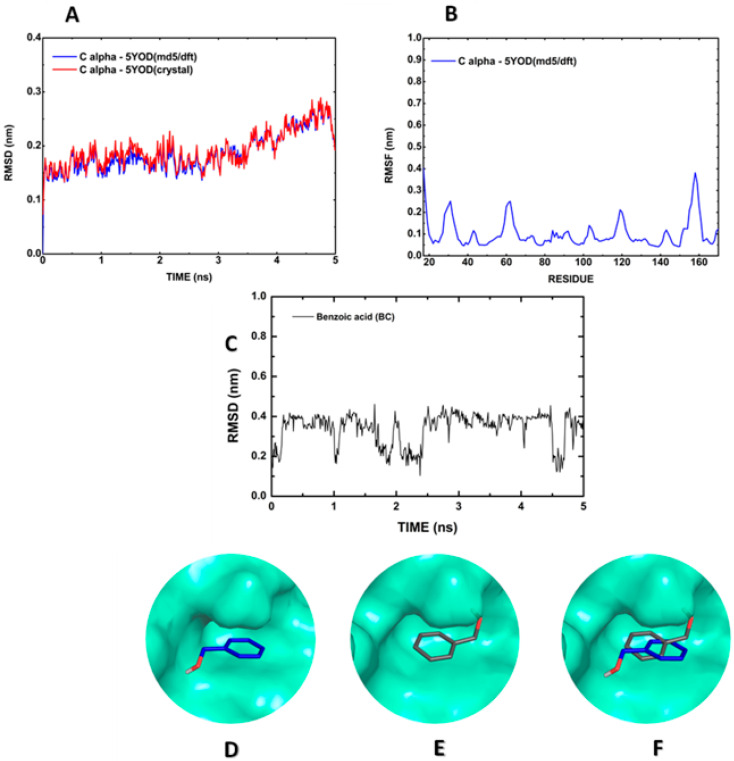
Molecular dynamics simulation of the BC:5YOD structure in water: (**A**) Cα RMSD; (**B**) Cα RMSF; (**C**) RMSD of BC relative to the protease; (**D**) BC position in the co-crystallized structure; (**E**) BC at 3 ns of the MD; (**F**) superposition of the co-crystallized BC position, and the MD final position.

**Figure 4 ijms-23-10088-f004:**
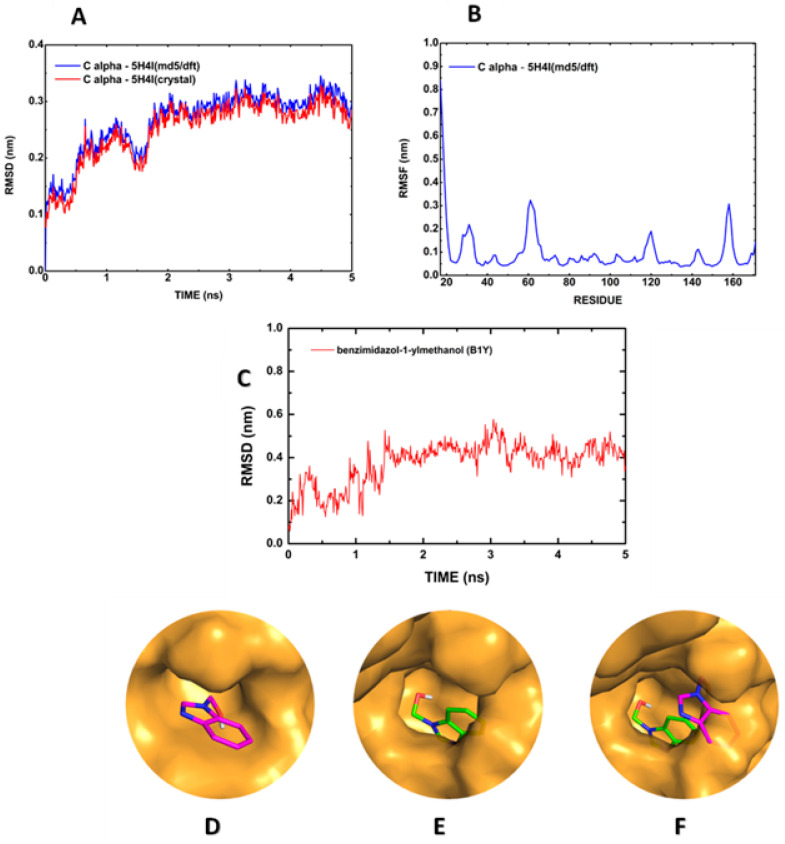
Molecular dynamics simulation of the B1Y:5H4I structure in water: (**A**) Cα RMSD; (**B**) Cα RMSF; (**C**) RMSD of B1Y relative to the protease; (**D**) B1Y position in the co-crystallized structure; (**E**) B1Y at 3 ns of the MD; (**F**) superposition of the co-crystallized B1Y position, and the MD final position.

**Figure 5 ijms-23-10088-f005:**
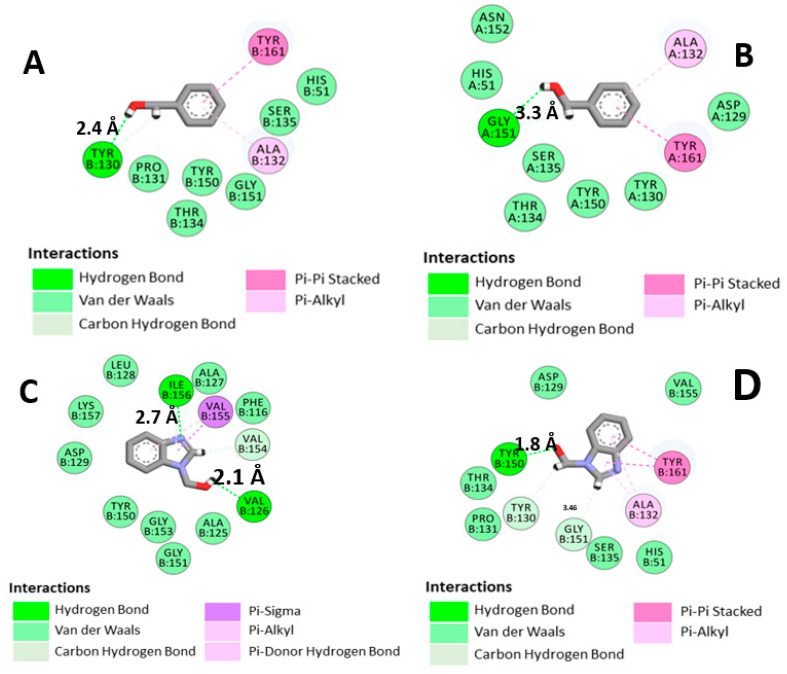
Two-dimensional diagrams of the main interactions of BC and B1Y with the Zika virus protease: (**A**) 5YOD (md5); (**B**) 5YOD (em); (**C**) 5H4I (md5); (**D**) 5H4I (em).

**Figure 6 ijms-23-10088-f006:**
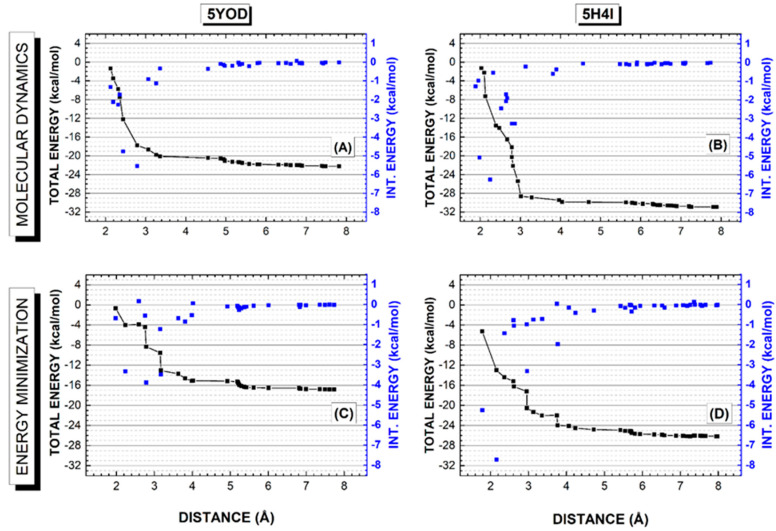
Quantum biochemistry calculations for the PDB structures 5YOD and 5H4I modified after a 5-ns molecular dynamics simulation (top) and after classical energy minimization (bottom). (**A**,**C**) 5YOD interaction energy (black line + scatter) and the interaction energies of the individual residues included as the distance from the ligand is increased (blue scatter and blue scale right axis). (**B**,**D**) 5H4I interaction energy (black line + scatter) and the interaction energies of the individual residues included as the distance from the ligand is increased (blue scatter and blue scale right axis).

**Figure 7 ijms-23-10088-f007:**
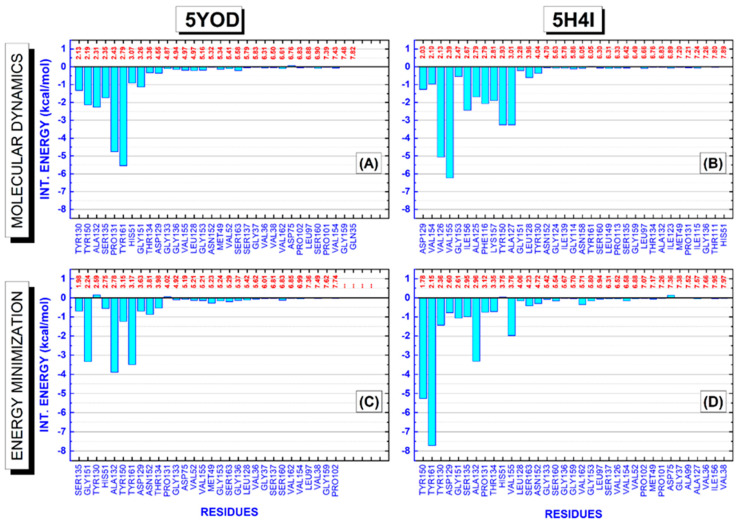
The binding site, interaction energy, and residue domain (BIRD) panels show the MFCC protein–ligand interaction energies with their respective distances to the ligand. (**A**,**C**) 5YOD (md5/dft) and 5YOD (em/dft) structures, respectively. (**B**,**D**) 5H4I (md5/dft) and 5H4I (em/dft) structures, respectively.

**Figure 8 ijms-23-10088-f008:**
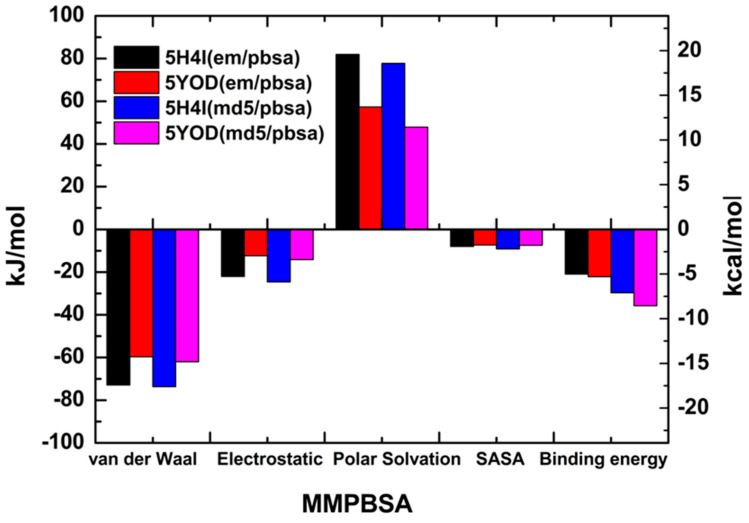
MM-PBSA-calculated energies for the em and md5 structures obtained from the PDB files 5H4I and 5YOD.

**Figure 9 ijms-23-10088-f009:**
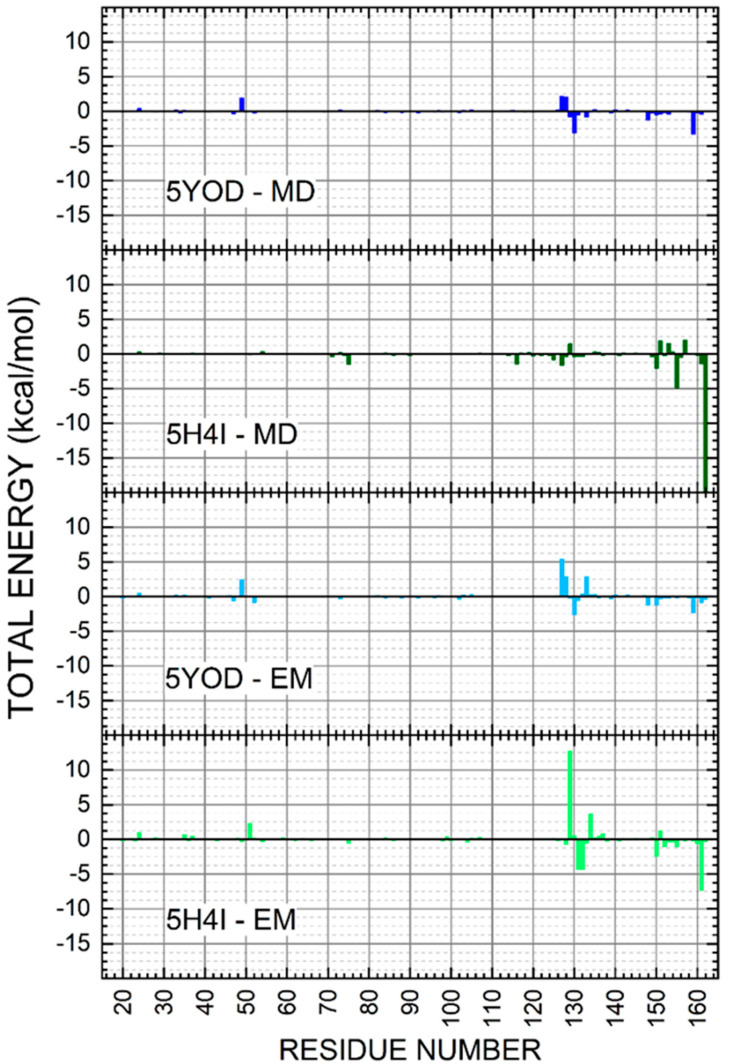
Energy contributions per amino acid residue for the 5YOD (md5/pbsa), 5YOD (em/pbsa), 5H4I (md5/pbsa), and 5H4I (em/pbsa) systems.

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
