# Peer review of "Quantum Biochemistry and MM-PBSA Description of the ZIKV NS2B-NS3 Protease: Insights into the Binding Interactions beyond the Catalytic Triad Pocket"

_ijms, 2022, doi:10.3390/ijms231710088_

Round 1

Reviewer 1 Report

The manuscript entitled “Quantum biochemistry and MM-PBSA description of the ZIKV NS2B-NS3 protease: insights into the binding interactions beyond the catalytic triad pocket” by the authors Valdir Ferreira de Paula Junior, Mauricio Fraga van Tilburg, Pablo Abreu Morais, Francisco Franciné Maia Júnior, Maria Izabel Florindo Guedes, Ewerton Wagner Santos Caetano, Valder Nogueira Freire is quite interesting and contemporary, applying state-of-the-art methods of the structure-based design with relevant post-processing analyses. Nevertheless there are some significant flaws:

Why were the ligands protonated at pH 4.6 instead of at physiological pH?

Molecular dynamic simulations should be performed in saline in order to mimic the real environment as much as possible.

Additionally the following major revisions should be considered:

The the protein Cα-RMSDs and RMSFs should be presented in order to evaluate the stability of the complex. A longer MD simulation is recommended.

Detailed docking protocol should be added, including exhaustiveness and energy range; rigid or flexible BS. In case the latter is applied, the flexible aminoacids (aa) should be denoted. The RMSD values of the re-docked ligands relative to the crystallographic structures should be given for protocol validation.

The positions of the aa of the catalytic triad within the BS should be clarified and could be presented on Figure 1A, together with the positions P1, P1’-P2 and connection area S1.

There are some minor revisions to be considered:

It would be more informative if the length of the hydrogen bonds are given in the discussion of Figure 4 instead of the distance to the aa.

Please denote the software used for the visualization of the intermolecular interactions of the studied complexes (Figure 4).

Figure 8 should be enlarged.

The annotation of Figure 2 should be below it.

Despite the above revisions for consideration the study is significant and I would recommend for the authors to improve the current version of the manuscript and re-submitted it for publication in the International Journal of Molecular Sciences.

Author Response

Dear Dr.
We greatly appreciate the evaluation of our article by the reviewers. The suggestions have been taken into account and included in the revised text. Please see the attachment.
Kind regards,

Reviewer 2 Report

This paper presents a modeling study of the protease complex necessary for replication of the Zika virus. Overall, the paper is very well-written. The figures are thoughtfully prepared, in general, and the presentation of methods, data, and interpretations are clear and concise. The insight resulting from the authors' analysis is somewhat limited, as they note in the conclusion paragraph, but the study is of certainly sufficient import to warrant publication, and this journal is an appropriate one for such a study, in my view. I have three minor criticisms of the manuscript that I would be happy to see addressed in final edits. 1. Figure 1A does not do a good job of clearly illustrating the structural differences discussed in the introduction. The difference in binding poses is clear, but the authors mention amino acid rearrangements that are of interest. It would be nice to see a more detailed view of the structures that highlights these differences. 2. In the first paragraph of the Results and Discussion, the authors mention regions of the binding pocket (S1, P1, etc.), but they offer no discussion of these regions anywhere in the manuscript. To any reader unfamiliar with such designations, this important point will be completely lost. I suggest the authors include a brief description of these terms before addressing them. Perhaps an illustration of the binding pocket illustrating these regions would be a helpful addition to Figure 1. 3) Similarly, the authors use abbreviations in the Results and Discussion section that have not yet been defined. For example, BC is not explained until the Materials and Methods section. This should be fixed to improve the clarity of the manuscript. 

Author Response

(The authors gave the same response as above.)

Round 2

Reviewer 1 Report

The authors made some revisions of the manuscript but there are missing comments on the significant flaws:

Why were the ligands protonated at pH 4.6 instead of at physiological pH?

Molecular dynamic simulations should be performed in saline in order to mimic the real environment as much as possible.

The protein Cα-RMSDs and RMSFs are also omitted.

For these reasons I am not convinced that this article is suitable to be published in the International Journal of Molecular Sciences.

Author Response

Dear Dr.

Kind regards,
Dr. Valdir Ferreira

Round 3

Reviewer 1 Report

The article is ready to be published.